# Immunohistochemical Characteristics of the Human Carotid Body in the Antenatal and Postnatal Periods of Development

**DOI:** 10.3390/ijms22158222

**Published:** 2021-07-30

**Authors:** Dmitry Otlyga, Ekaterina Tsvetkova, Olga Junemann, Sergey Saveliev

**Affiliations:** Research Institute of Human Morphology, 117418 Moscow, Russia; tsvetkovakatya@mail.ru (E.T.); ojunemann@yandex.ru (O.J.); braincase@yandex.ru (S.S.)

**Keywords:** carotid body, ontogeny, human development, tyrosine hydroxylase, endocrine function

## Abstract

The evolutionary and ontogenetic development of the carotid body is still understudied. Research aimed at studying the comparative morphology of the organ at different periods in the individual development of various animal species should play a crucial role in understanding the physiology of the carotid body. However, despite more than two centuries of study, the human carotid body remains poorly understood. There are many knowledge gaps in particular related to the antenatal development of this structure. The aim of our work is to study the morphological and immunohistochemical characteristics of the human carotid body in the antenatal and postnatal periods of development. We investigated the human carotid bodies from 1 embryo, 20 fetuses and 13 adults of different ages using samples obtained at autopsy. Immunohistochemistry revealed expression of βIII-tubulin and tyrosine hydroxylase in the type I cells and nerve fibers at all periods of ontogenesis; synaptophysin and PGP9.5 in the type I cells in some of the antenatal cases and all of the postnatal cases; 200 kDa neurofilaments in nerve fibers in some of the antenatal cases and all of the postnatal cases; and GFAP and S100 in the type II cells and Schwann cells in some of the antenatal cases and all of the postnatal cases. A high level of tyrosine hydroxylase in the type I cells was a distinctive feature of the antenatal carotid bodies. On the contrary, in the type I cells of adults, the expression of tyrosine hydroxylase was significantly lower. Our data suggest that the human carotid body may perform an endocrine function in the antenatal period, while in the postnatal period of development, it loses this function and becomes a chemosensory organ.

## 1. Introduction

More than 277 years have passed since the discovery of the carotid body. However, issues surrounding the evolutionary and ontogenetic development of the organ remain unresolved. Although the carotid body was first assigned as a part of the unitary paraganglionic system by Alfred Kohn [1], Fernando de Castro established the sensory innervation of the organ [2,3]. Since then, the carotid body has been considered only to be a chemoreceptor organ by most researchers [4].

However, a number of researchers have continued to deny the generally accepted chemosensory function of the carotid body cells. Their position was based on ultrastructural characteristics of the organ, such as the presence of a large number of efferent synapses on the type I cells [5,6,7,8,9] and the absence of the polarity intrinsic to sensory cells [9].

The studies of Nina Smitten on the comparative histology and embryology of the carotid body are especially interesting. Smitten suggested the carotid body to be nothing more than rudimentary chromaffin tissue that had lost its endocrine functions due to the appearance of a well-developed adrenal medulla and reduction of the branchial vessels in mammals [10].

However, further studies refuted the thesis of a rudimentary character of the carotid body and proved that the type I cells show chemoreceptor function [11,12,13,14]. This also led to the rejection of the key point of Smitten’s hypothesis of the phylogenetic unity of the adrenal medulla, the carotid body and the organ of Zuckerkandl.

Although the similarities between these organs are well known, the mainstream theory of the evolution of the carotid body suggests that the homologs of the carotid body are the innervated neuroepithelial cells of the gills in fish [15,16]. This view of the evolution of the carotid body has meant that, for a long period of time, the carotid body has not been considered to perform endocrine functions together with the adrenal medulla and the organ of Zuckerkandl. Only recently, Hockman et al. demonstrated that the neuroepithelial cells of the gills are not homologs of the type I cells of the carotid body [17]. Their investigation shows that catecholaminergic cells associated with branchial arch blood vessels are more likely to be homologs of the carotid body in mammals, confirming Smitten’s views on the evolution of chromaffin tissue. 

Difficulties in the interpretation of the functions of the carotid body show the importance of comparative anatomic methods of investigation. In this regard, the carotid body of the human is of particular interest.

Although today there is an increased interest in this problem, characteristics of the human carotid body are still underinvestigated. There are many studies concerning the anatomy and histology of the carotid body, but there is still very little research on the molecular genetics and immunohistochemical features of this organ in humans [18,19,20,21]. In addition, the antenatal period of development of the human carotid body is also poorly studied.

This is in part due to the ethical and technical problems with obtaining human material. Additionally, autolytic changes in the organ lead to various artifacts, such as dark and pyknotic cells [22,23], making it difficult to conduct and interpret the results of immunohistochemical investigations [24].

In our previous work, we showed that the carotid body has immunohistochemical characteristics similar to those of the organ of Zuckerkandl and the adrenal medulla in the antenatal period [25]. The aim of this study is to investigate and compare features of the human carotid body in the antenatal and postnatal periods.

## 2. Results

### 2.1. Comparison of the Human Carotid Body Structure in the Antenatal and Postnatal Periods 

In the antenatal and postnatal periods, all studied carotid bodies were located in the area of bifurcation of the carotid artery. They consisted of two cell types, grouped into glomeruli: a group of oval or round type I cells surrounded on the periphery by type II fusiform cells.

In Case 1 (8 PCW embryo), total serial sections allowed assessment of the comparative size of the organ in relation to other structures. The maximum cross-sectional area of the left carotid body was 6.7 times greater than the cross-sectional area of the left internal carotid artery, and the cross-sectional area of the right carotid body was 7.1 times larger than the cross-sectional area of the right internal carotid artery. Taking the ratio of the largest organ size to the diameter of the internal carotid artery, the size of the left carotid body was 3.3 times the diameter of the left internal carotid artery, and the right carotid body was 4.04 times the diameter of the right internal carotid artery. Thus, by the 8 PCW, the carotid body is a relatively large organ, exceeding the size of the main vessels of the embryo.

As the gestational age increased, the morphology of the organ changed as follows. In Case 1, the type I cells were mostly round or oval with a narrow, almost imperceptible rim of basophilic cytoplasm (Figure 1A). In Case 2, there was a prominent increase in the volume of the cytoplasm (Figure 1B). In the rest of the fetuses, an increase in the volume of the cytoplasm of type I cells was found, which also developed mild eosinophilia. The cells’ cytoplasm was foamy and disintegrated into lumps (an artifact of formalin fixation) (Figure 1C,D).

The presence of so-called “pyknotic” cells was noted in Case 1 (8 PCW), and subdivision into “dark” and “light” cell subtypes was observed from 18 to 19 GW. This was found to be the result of autolysis, as we have established previously [24].

In Cases 1 and 2, the glomeruli had barely discernible borders, as the type II cell processes were very thin. In addition, the glomeruli in these two cases did not form lobules (Figure 1A,B). Well-defined lobules consisting of groups of glomeruli were first noted in Case 3 at a gestational age of 13–14 (Figure 1C).

As the gestational age increased, the lobules became more distinct, surrounded by fusiform cells along the perimeter and separated from each other by thin layers of sparse connective tissue, which consisted mainly of fusiform cells (Figure 1D).

The carotid bodies in Cases 1 and 2 were surrounded by a thin rim of 1–3 layers of elongated mesenchymal cells. However, the formation of a connective tissue capsule was not observed. In Fetuses 3–21, the formation of a capsule around the carotid body was also not observed.

On the periphery of the organ, there was local rarefaction of the connective tissue around the glomeruli. This connective tissue consisted of rare collagen fibers and spindle-shaped fibroblast-like cells. We also noted a congestion of denser collagen fibers in several layers and the presence of a large number of fusiform cells. However, this formation is not a true capsule, despite resembling it in structure. The structure was discontinuous, with areas of rarefaction of collagen fibers. Thin-walled veins filled with erythrocytes were sometimes found around the carotid body.

The carotid bodies of adults resembled fetal organs in histological structure but had much more connective tissue and more prominent lobules (Figure 2A,B). In all adult carotid bodies, the type I cells also had oval or rounded nuclei. Their cytoplasm was light and eosinophilic and was often fragmented (formalin fixation artifact). As in the antenatal period, there were three cell subtypes found in adults: light, dark and pyknotic (an artifact of autolysis).

Type II cells were spindle shaped and had elongated nuclei, surrounding type I cells on the periphery. However, these cells were hard to distinguish from stromal and Schwann cells, which had similar morphology and were located close to the glomeruli. 

In Case 22 (the 24-year-old woman), the organ was surrounded by fibrous connective tissue along the entire perimeter (Figure 2A). At the cranial and caudal poles of the organ, this tissue had well-defined borders with layers of adipose tissue. From the edge adjacent to the internal carotid artery, fibrous connective tissue passed imperceptibly into the adventitia of the artery. On the opposite edge, it had a more distinct border with looser connective tissue. The denser fibrous connective tissue separated the organ into lobules and carried large numbers of blood vessels and nerves. 

In other cases (23–34), we found a significant increase in connective tissue in the area of bifurcation of the common carotid artery. Connective tissue grew thickly between the lobules of the carotid body and seemed to grow into previously single lobules, dividing them into smaller ones (Figure 2B).

### 2.2. Immunohistochemical Study of the Human CAROTID Body 

In our previous study, we showed that some markers used in immunohistochemistry were especially sensitive to autolysis (neurofilaments and GFAP) [24]. In the current study, we found that, besides these markers, PGP9.5, synaptophysin and S100 were also unstable in the antenatal material, showing a significant decrease in the intensity of staining up to its absence. However, these markers were stable in adults (Table 1). Taking this into consideration, we could use only antibodies to βIII-tubulin and tyrosine hydroxylase for quantitative assessment of differences between carotid bodies, while the rest of the markers could be used for only qualitative assessment. 

In the antenatal and postnatal periods, the distribution pattern of markers in various structures of the carotid body was similar to that seen in other mammals (Table 2, Figure 3, Figure 4 and Figure 5).

However, when comparing the expression of tyrosine hydroxylase, significant differences between the antenatal and postnatal periods were observed (Table 3, Figure 5 and Figure 6). As early as 8 PCW, the type I cells demonstrated strong expression of tyrosine hydroxylase. Moreover, in all cases, the TH/βIII ratio was rather high, with a median value of 0.8. A minimum of 0.51 was observed in Case 17, and the highest values were seen in Cases 1 and 5 (0.92 and 0.93, respectively). There was no correlation between gestational age and TH/βIII ratio. In contrast to the antenatal period, in postnatal material the type I cells showed a weak reaction with tyrosine hydroxylase. The TH/βIII ratio was low, with a median value of 0.09. Only a few cells showed a distinct reaction to tyrosine hydroxylase. There was a greater number of TH-positive cells only in Cases 25 and 30. A minimum of 0.01 was observed in Case 34, and a maximum of 0.49 was noted in Case 25.

As shown before, it was difficult to trace the nerve fibers of the carotid body, as they express very similar markers to the cells of the organ itself. The only marker expressed specifically by nerve fibers was neurofilaments. However, due to its instability, this marker was unsuitable to use for antenatal development studies. 

In the antenatal period, there were positive reactions with βIII-tubulin and tyrosine hydroxylase in the nerve fibers. However, in Cases 1 (8 PCW) and 2 (10 PCW), the carotid body cells were located so close to each other that it was impossible to trace the nerve fibers inside the organ. From 13 to 14 GW (Case 3), the lobules appear and are separated from each other by loose connective tissue so that tracing of nerve fibers becomes possible. The nerve fibers enter the organ and pass through the interlobular space, enlacing the lobules with a dense network.

In the postnatal period, it was also possible to trace the nerve fibers inside the lobules. These NF-positive nerve fibers entered the glomeruli and formed intermediate enlargements and end buds. They were often observed in the areas of contact with type I cells. Individual variability in the density of the nerve fibers was significant.

## 3. Discussion

The experiments of de Castro and Heymans resulted in the scientific community being convinced that the carotid body was a chemoreceptor organ, and the hypothesis of endocrine function suggested by early researchers was rejected.

Our results demonstrate that even at the earliest developmental stages (8 PCW), the carotid body is already a relatively large structure, approximately seven times the area of the cross-section of the internal carotid artery, which is much larger than in adults. In addition, from the earliest gestational ages, the type I cells already synthesize βIII-tubulin and tyrosine hydroxylase intensively, suggesting that the carotid body has a significant synthetic activity.

Our results agree with the study by Korkala and Hervonen, who found that catecholamines were present in the carotid body cells of the 8 PCW embryo, using formalin-induced fluorescence [26]. However, the crown-rump length (CRL) of the embryo in their investigation was 28 mm, while in our study it was 24–25 mm. Thus, the embryo we examined was of an earlier stage.

With the increase in gestational age, the volume of the type I cell cytoplasm grew and acquired pale eosinophilia. From 13 to 14 GW, lobules formed by glomeruli could be detected, slightly earlier than observed by Scraggs et al. [27]. Later, only an increase in the volume of the organ was noted.

A high TH/βIII ratio was noted at all stages of prenatal development, which may indicate a high level of catecholamine synthesis by the carotid body during the entire antenatal period of ontogenesis.

At the earliest developmental stages, it was difficult to trace the nerve fibers inside the organ due to the dense arrangement of cells. However, the presence of a well-developed nerve network at 13–14 GW suggests that the organ had acquired rich innervation at an earlier stage.

A low tyrosine hydroxylase level in the type I cells of the carotid bodies is a distinctive feature of the postnatal period. According to the literature, in four-month-old infants, the number of TH-positive cells is already relatively low [18]. Ortega-Sáenz et al. reported that the number of TH-positive cells was surprisingly low throughout postnatal development of the human carotid body [21]. This feature distinguishes the carotid bodies of adults and children not only from the organs of fetuses and embryos but also from the carotid body of rats and mice, which is also characterized by a high content of tyrosine hydroxylase, as evidenced by data in the literature [18,20,28,29].

However, this does not suggest a higher level of the catecholamines themselves in the type I cells of the rat carotid body. While producing plenty of tyrosine hydroxylase, type I cells show a very low level of catecholamines, which indicates a slowdown in synthesis at subsequent stages of biochemical transformations. Nevertheless, there is also a decrease in the production of tyrosine hydroxylase after birth in rats [30]. At the same time, a number of authors note that, at the moment of birth, the sensitivity of type I cells is minimal, whereas later it significantly increases [31,32,33].

While considering the carotid body in association with the organ of Zuckerkandl and the adrenal medulla, we encounter an interesting phenomenon. In the embryonic and fetal periods, the human carotid body contains a large number of TH-positive cells, and its immunohistochemical characteristics are almost identical to those of the organ of Zuckerkandl and the adrenal medulla [25].

Our previous study demonstrates that, in the embryonic and fetal periods, the carotid body is already a fairly large structure, whereas the adrenal medulla has only small scattered groups of chromaffin cell precursors, which are mostly immature. In addition, we could not find any adrenal medulla cells on hematoxylin–eosin-stained sections in 8 PCW embryos without the use of immunohistochemistry, which showed only a few TH-positive cells [25].

This may signify that, during the antenatal period, the carotid body performs an endocrine function, along with the organ of Zuckerkandl. Both organs synthesize catecholamines, thereby compensating for the insufficiency of the adrenal medulla function at this stage of development.

It is known that the adrenal medulla remains immature even after birth, and the organ of Zuckerkandl probably undertakes its function, reaching its maximum by the age of three. Between three and five years of age, the organ of Zuckerkandl undergoes involution, with the simultaneous growth and maturation of the adrenal medulla [34].

Hence, Smitten’s theory of phylogenesis of the chromaffin tissue is well founded. In the evolution from the cyclostomes to mammals, the chromaffin tissue of the branchial arteries undergoes reduction alongside the development of the adrenal medulla [10]. Thus, the chromaffin tissue of the branchial arteries loses its endocrine functions, which are then performed by the adrenal medulla alone. However, Smitten’s work does not consider the possibility of chemoreceptor specialization of the carotid body.

We suggest that there is a similar phenomenon in the ontogenesis of humans. The type I cells of the human carotid body reduce their synthesis of tyrosine hydroxylase almost immediately after birth. Then, over the first two years of postnatal development, the organ of Zuckerkandl undergoes involution, and the adrenal medulla undertakes the function of catecholamine synthesis. At the same time, the carotid body switches over from endocrine to chemoreceptor function. Thus, the process of individual development of the paraganglia system partially repeats the phylogenesis of chromaffin tissue.

In addition, further evidence of the unity of the adrenal medulla and the carotid body is that both organs react similarly to acute and chronic hypoxia, with the release of catecholamines and hyperplasia of chromaffin cells, respectively [35,36]. The fact that there is an increase in the synthesis of catecholamines in both organs in response to chronic hypoxia suggests that both have an endocrine function.

However, the most exciting result is related to the sensory innervation of the carotid body. In most studies, the presence of efferent and reciprocal synapses on the axon terminals of sensory neurons of glossopharyngeal ganglion was observed [6,7,8,9,37,38,39,40]. Nevertheless, the majority of researchers interpreted their presence as functioning in chemoreceptor function regulation.

The decrease in the chemoreceptor response of type I cells to electric stimulation of glossopharyngeal nerve has been reported, and this effect is mediated by the release of catecholamines [41]. All explanations of this phenomenon from the standpoint of chemoreceptor function alone seem insufficient.

We suggest that this phenomenon could be explained by the presence of the axon reflex in the carotid body. Thus, according to Zavarzin, sensory neurons can also simultaneously perform an efferent function. This explains, for example, the skin-vascular reaction in response to irritation [42].

It is likely that a similar process happens in the carotid body. When detecting slight changes in the gases of the blood, it does not make sense to use the mechanisms of the sympathoadrenal system. However, under severe hypoxia, the type I cells must stimulate the sensory fibers of the sinus nerve more strongly, releasing more acetylcholine and ATP as mediators. In response, sensory fibers not only transmit a signal to the brainstem but also in the opposite direction to the type I cells through their own efferent synapses. This results in the enforced release of catecholamines, including dopamine, by the type I cells. This can explain the results of the experiments by Hellström, who observed a decrease in the release of dopamine and noradrenaline by the type I cells in response to hypoxia in the 14 days after sinus nerve transection [43].

This hypothesis also perfectly explains the observations of Zapata, who noted a biphasic effect of dopamine. Dopamine application firstly depressed the frequency of sinus nerve discharges, but after repeated injections applied at short intervals, the inhibitory effect was replaced by excitation [44]. It is likely that the initial excitation with acetylcholine leads to an increase in the frequency of sinus nerve discharges. If the homeostasis does not return to normal, type I cells begin to release dopamine fractionally, both on their own and under the influence of the sinus nerve. Dopamine leads to an increase in the impulse frequency of the sinus nerve in the second phase. Moreover, it may be released into the bloodstream, resulting in systemic effects. However, these effects are not strong in adults due to the reduction of the endocrine function of the carotid body. 

In conclusion, the presence of a large amount of efferent and reciprocal synapses alongside afferent synapses becomes much more easily explainable. Thus, although type I cells of the adult carotid body partially lose their endocrine functions, they perform these functions during the embryonic and fetal periods. This agrees with our results, which show the presence of a large amount of tyrosine hydroxylase in embryonic and fetal cells, and with the discovery of catecholamines in work by Korkala [26]. For performing these endocrine functions, the organ has a rich efferent innervation, which remains in the postnatal period as a vestigial structure. In addition, research by Hervonen confirms the well-developed innervation of type I cells of the carotid bodies of fetuses in the second trimester of intrauterine development [8], which indicates the presence of developed nervous regulation of organ functions.

## 4. Materials and Methods

Autopsy material was obtained from GBUZ MO MONIIAG, S.S. Yudin City Clinical Hospital, FSBI “National medical research center for obstetrics, gynecology and perinatology named after academic V.I. Kulakov”, Sechenov University, City Clinical Hospital 31. In our study, material from 1 embryo, 20 fetuses and 13 adults was used (Table 4). 

While obtaining samples, gender, gestational age, clinical diagnosis of the mother and fetus and causes of termination of pregnancy and of fetal deaths were recorded. If clinical data were insufficient or absent, fetal age was determined according to the criteria developed by Milovanov and Saveliev [45]. In adults, gender, age, clinical and pathologic diagnoses were recorded. Persons with severe chronic hypoxia and chronic respiratory failure were excluded.

Embryo 1 was totally fixed in 10% buffered formalin, paraffin-embedded and sectioned fully. For the rest of the fetuses, dissection of the carotid arteries in the area of bifurcation with surrounding connective tissue was performed. In adults, the carotid arteries in the bifurcation area were dissected together with adjacent tissues, then the carotid bodies were isolated from the surrounding tissues. Carotid bodies were fixed in 10% buffered formalin.

Tissue specimens were dehydrated using a standard tissue protocol with IsoPrep BioVitrum (St. Petersburg, Russia) and embedded in Histomix BioVitrum (St. Petersburg, Russia). Total serial sections (thickness = 6 μm) were taken using a Leica RM2245 microtome (Wetzlar, Germany). Every 20th section was taken, deparaffinized and stained routinely with hematoxylin and eosin. The sections were examined under a Leica DM2500 light microscope (Wetzlar, Germany).

For immunohistochemical studies, the largest tissue sections were chosen. They were deparaffinized, rehydrated and treated with 3% H_2_O_2_ peroxide solution for 10 min to block endogenous peroxidase. Then, the sections were treated with Ultra V Block (Thermo Fisher Scientific, Waltham, MA, USA) for 5 min and boiled in citrate buffer (pH 6.0) for up to 30 min for antigen retrieval. Then, the sections were incubated with primary antibodies for 60 min at room temperature (Table 5). The UltraVision Quanto Detection System kit by Thermo Fisher Scientific (Waltham, MA, USA) was used as the detection system.

Superior cervical ganglion and human brain sections were used as the positive control. Sections incubated with phosphate buffer solution instead of primary antibodies served as the negative control.

The sections were photographed with a LOMO TCA-9.0 digital camera (LOMO, St. Petersburg, Russia) at 100× for the morphometric analysis. The digital images were saved in JPEG and TIFF formats, matched for brightness and contrast using Adobe Photoshop CC 2019 (Adobe Systems, Inc., San Jose, CA, USA). Morphometry was performed using the program ImageJ 1.52 n (Bethesda, MD, USA).

To analyze the levels of tyrosine hydroxylase synthesis, we used the TH/βIII coefficient, which was calculated as the ratio of the sample area stained with tyrosine hydroxylase to the area stained with βIII-tubulin. The areas were automatically calculated using the ImageJ 1.52 n software (Bethesda, MD, USA), and the ratio was calculated using Microsoft Excel 2019.

Statistical significance of the results was assessed using Statistica 13.5 software (TIBCO Software Inc., Palo Alto, CA, USA). The type of distribution was determined using the Kolmogorov–Smirnov criterion. As the distribution was different from normal, the significance of differences was assessed using the Mann–Whitney test. The differences were considered significant at *p* < 0.05.

## 5. Conclusions

It was shown that by the eighth postconception week, the carotid body is a large structure, actively synthesizing βIII-tubulin and tyrosine hydroxylase. Throughout the antenatal and postnatal periods, the type I cells were positive for βIII-tubulin, PGP9.5 and synaptophysin, and the type II cells and Schwann cells were positive for GFAP and S100. Intraorgan nerve fibers showed positive reactions to βIII-tubulin, PGP9.5, tyrosine hydroxylase and 200 kDa neurofilaments.

A high level of tyrosine hydroxylase in the type I cells was a distinctive feature of the antenatal carotid bodies. On the contrary, in the type I cells of adults, the expression of tyrosine hydroxylase was significantly lower. 

Our data suggest that, in the antenatal period, the human carotid body may perform an endocrine function, while in the postnatal period of development, it loses this function and becomes a chemosensory organ.

## Figures and Tables

**Figure 1 ijms-22-08222-f001:**
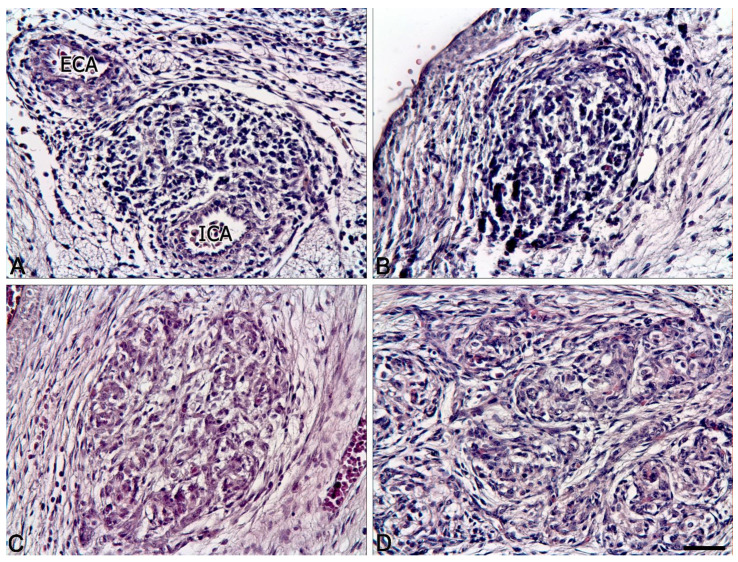
The human carotid body in the antenatal period: (**A**) 8 PCW embryo; (**B**) 10 PCW prefetus; (**C**) 13–14 GW fetus; (**D**) 19–20 GW fetus. ECA, external carotid artery, ICA, internal carotid artery. H&E stain. Bar = 50 μm.

**Figure 2 ijms-22-08222-f002:**
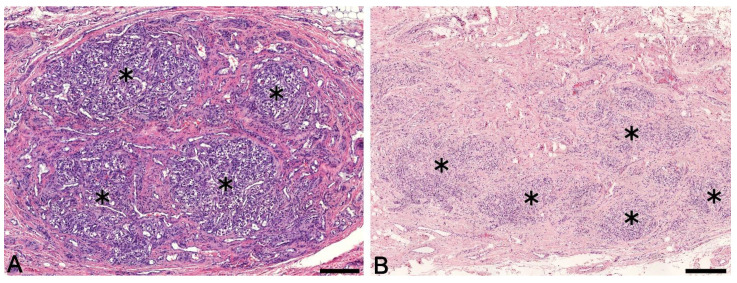
The human carotid body in the postnatal period. (**A**) The 24-year-old woman. (**B**) The 86-year-old woman. Asterisks represent lobules. H&E stain. Left bar = 200 μm; right bar = 400 μm.

**Figure 3 ijms-22-08222-f003:**
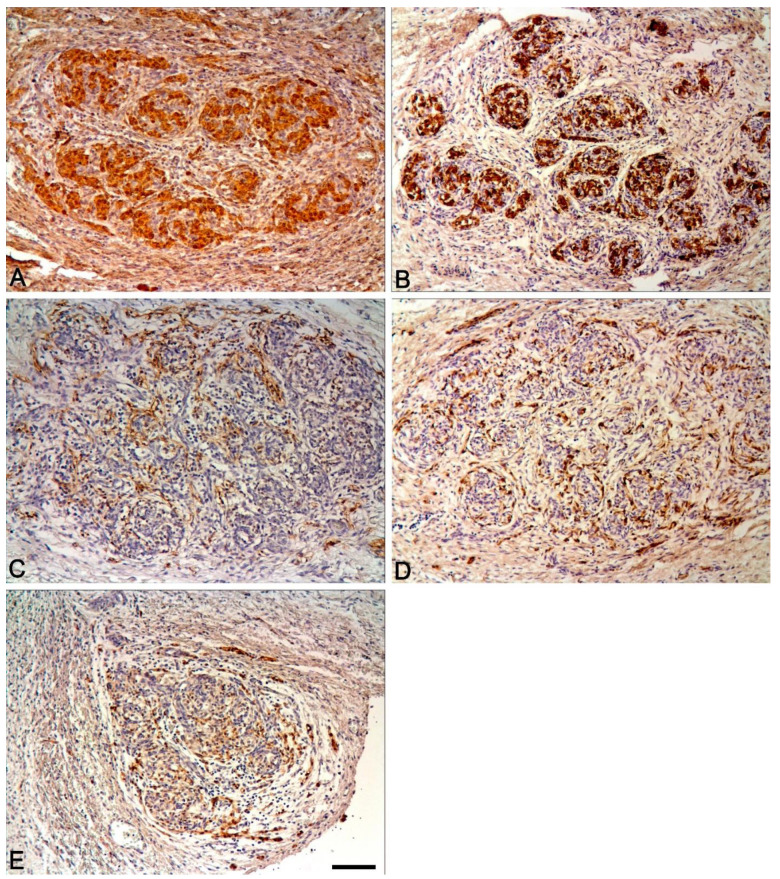
The human carotid body in the antenatal period. Expression of PGP9.5 (**A**), synaptophysin (**B**), neurofilaments (**C**), GFAP (**D**) and S100 (**E**). Nuclei stained with hematoxylin. Bar = 100 μm.

**Figure 4 ijms-22-08222-f004:**
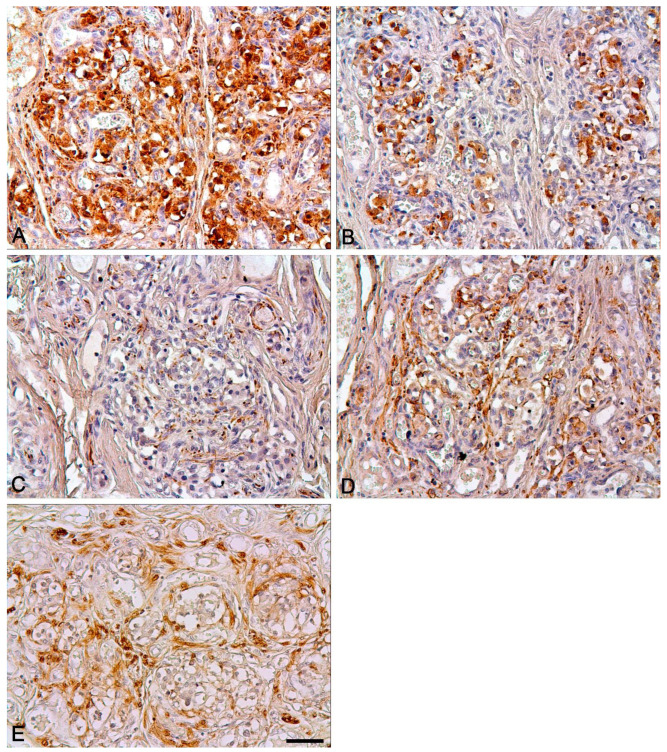
The human carotid body in the postnatal period. Expression of PGP9.5 (**A**), synaptophysin (**B**), neurofilaments (**C**), GFAP (**D**) and S100 (**E**). Nuclei stained with hematoxylin. Bar = 50 μm.

**Figure 5 ijms-22-08222-f005:**
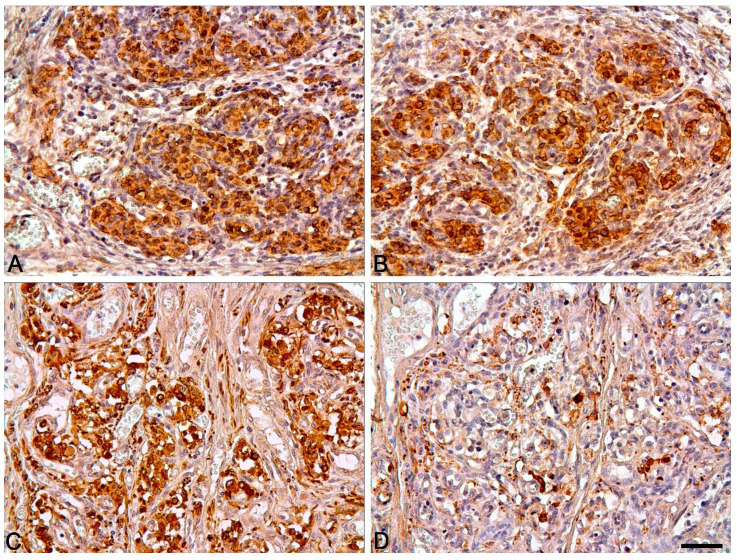
Strong expression of βIII-tubulin (**A**) and tyrosine hydroxylase (**B**) in the carotid body of the 19–20 GW fetus. Strong expression of βIII-tubulin (**C**) and low expression of tyrosine hydroxylase (**D**) in the carotid body of the 78-year-old woman. Nuclei stained with hematoxylin. Bar = 50 μm.

**Figure 6 ijms-22-08222-f006:**
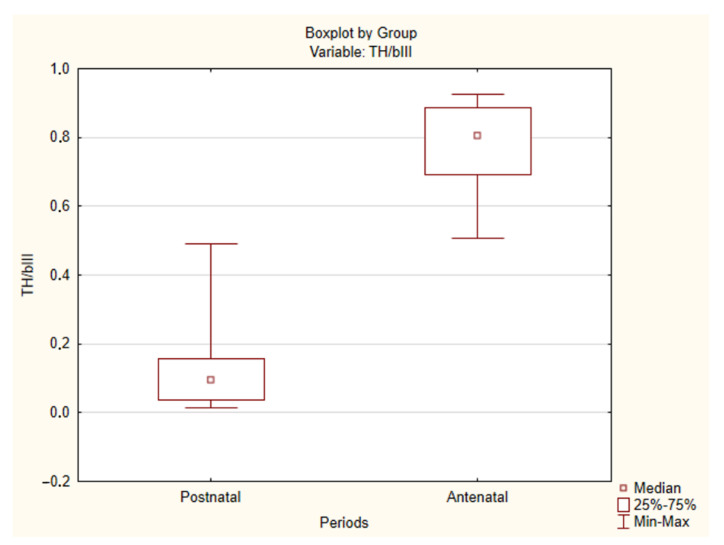
TH/βIII ratio in the antenatal and postnatal human carotid body. Statistically significant differences (*p* < 0.05).

**Table 1 ijms-22-08222-t001:** Expression of markers in different samples of the human carotid body.

Case No	βIII-Tubulin	PGP9.5	NF 200 kd	TH	Syn	GFAP Ab-4	S100
Antenatal							
1	+			+			
2	+			+			
3	+			+			
4	+	+		+	+		
5	+	+		+	+		
6	+	+	+	+	+		
7	+	+		+	+		+
8	+	+	+	+	+	+	
9	+	+	+	+	+	+	
10	+	+		+	+		
11	+	+		+	+		
12	+	+		+	+		
13	+	+		+	+		
14	+			+			
15	+			+			
16	+			+			
17	+			+			
18	+			+			
19	+	+		+	+	+	+
20	+	+		+	+	+	+
21	+	+		+	+		
Postnatal							
22	+	+	+	+	+	+	+
23	+	+	+	+	+	+	+
24	+	+	+	+	+	+	+
25	+	+	+	+	+	+	+
26	+	+	+	+	+	+	+
27	+	+	+	+	+	+	+
28	+	+	+	+	+	+	+
29	+	+	+	+	+	+	+
30	+	+	+	+	+	+	+
31	+	+	+	+	+	+	+
32	+	+	+	+	+	+	+
33	+	+	+	+	+	+	+
34	+	+	+	+	+	+	+

«+» Detection of marker expression.

**Table 2 ijms-22-08222-t002:** Expression of markers in different structures of the human carotid body.

Case No.	Type I Cells	Type II Cells	Schwann Cells	Nerve Fibers
βIII-tubulin	+			+
PGP9.5	+			+
NF 200 kd				+
TH	+			+
Syn	+			
GFAP Ab-4		+	+	
S100		+	+	

**Table 3 ijms-22-08222-t003:** TH/βIII ratio in the human carotid body.

Case No	TH/βIII Ratio	Case No	TH/βIII Ratio	Case No	TH/βIII Ratio
Antenatal period				Postnatal period	
1 ^L^	0.900731	12	0.70254	22	0.145537
1 ^R^	0.924528	13	0.854295	23	0.035849
2.	0.885407	14	0.624743	24	0.226393
3.	0.807854	15	0.535936	25	0.491257
4.	0.881466	16	0.646983	26	0.026529
5.	0.92536	17	0.507709	27	0.095764
6.	0.732308	18	0.691227	28	0.156351
7.	0.804473	19	0.892741	29	0.130611
8.	0.771863	20	0.845743	30	0.467732
9.	0.833736	21	0.910662	31	0.082004
10.	0.518871			32	0.058105
11 ^L^	0.70143			33	0.023186
11 ^R^	0.727811			34	0.014426

^L^ Left carotid body. ^R^ Right carotid body.

**Table 4 ijms-22-08222-t004:** Fetal and adult human ages.

Antenatal Period		Postnatal Period	
Case No.	Gestational Age	Case No.	Age, Years
1	8 PCW ^1^	22	24
2	10 PCW	23	87
3	13–14 GW ^2^	24	63
4	18–19 GW	25	69
5	19–20 GW	26	62
6	21–22 GW	27	78
7	17–18 GW	28	67
8	30 GW	29	86
9	30 GW	30	80
10	19–20 GW	31	79
11	19–20 GW	32	95
12	19–20 GW	33	68
13	16–17 GW	34	56
14	14–15 GW		
15	15–16 GW		
16	15–16 GW		
17	18–19 GW		
18	15–16 GW		
19	16 GW		
20	16 GW		
21	21 GW		

^1^ Postconception weeks. ^2^ Gestation weeks.

**Table 5 ijms-22-08222-t005:** Primary antibody characteristics.

No.	Antigen, Host Species, Supplier	Working Dilution
1	βIII-tubulin, rabbit polyclonal. Abcam (Cambridge, UK)	1:500
2	PGP9.5, murine monoclonal. Thermo Fisher Scientific (Waltham, MA, USA)	1:300
3	Neurofilaments 200 kDa, murine monoclonal. Merck (Darmstadt, Germany)	1:160
4	S100, rabbit polyclonal. Thermo Fisher Scientific (Waltham, MA, USA)	1:1200
5	Tyrosine hydroxylase, rabbit polyclonal. Abcam (Cambridge, UK)	1:160
6	GFAP Ab-4, rabbit polyclonal. Thermo Fisher Scientific (Waltham, MA, USA)	1:200–1:1000
7	Synaptophysin, murine monoclonal. Abcam (Cambridge, UK)	1:100

## Data Availability

Data is contained within the article.

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
