# Peer review of "Immunohistochemical Characteristics of the Human Carotid Body in the Antenatal and Postnatal Periods of Development"

_ijms, 2021, doi:10.3390/ijms22158222_

Round 1

Reviewer 1 Report

The manuscript by Otlyga and colleagues investigate the immunohistochemical characteristics of the human carotid body in the antenatal and postnatal periods of development. I found this work of great importance because there is a lack of knowledge about the morphological characteristics of the human carotid body and its alterations with age, sex, pathologies among others, it might also contribute to translational studies aiming to find therapeutics for diseases involving the dysfunction of the carotid body.

However, there are some points in the present work that deserve clarification and there is some important literature that is missing and deserves to be discussed. Also, some of the findings presented herein are not new – small % of TH cells in human carotid bodies as previously reported by Lazarov et al. 2009 (Resp Physiol Neurobiol) and by Ortega-Sanz et al. 2013 (J Physiol) and the decrease of TH cells with age (Ortega-Sanz et al. 2013).

Comments to the authors:

  • Have the authors considered to analyze left vs right carotid bodies. There are evidences that the left carotid body is bigger than the right one and this may be crucial for the analysis herein performed. Also, evidences suggest differences between females vs males. Could the authors include this information in the manuscript. Please see the manuscript by Ortega-Sanz et al. 2013 J Physiol - https://www.ncbi.nlm.nih.gov/pmc/articles/PMC3892469/. The results of the present manuscript must be discussed in the
  • Sympathetic-mediated diseases have shown to change CB size in humans with the carotid bodies of patients with diabetes mellitus, hypertension, and congestive heart failure being 20-25% larger relative to controls (Cramer et al. 2014 https://pubmed.ncbi.nlm.nih.gov/24156799/). Also, the majority of the carotid bodies evaluated in the present study were from very old patients probably under drug regimens. These 2 factors might interfere with the relative expression of TH and with the ratio TH/BetaIII-tubulin.
  • Ortega-Sanz et al. 2013 previously reported a decrease of TH expression with age in the human carotid bodies something that was already reported in rats (Conde et al. 2006, J Neurochem https://pubmed.ncbi.nlm.nih.gov/16899065/ and Sacramento et al. 2019 https://pubmed.ncbi.nlm.nih.gov/31426127/ )and in agreement with the decline of carotid body functionality with ageing , demonstrated by decreased hypoxic responses. Probably these literature deserves to be discussed.

Author Response

Thank you for your Review Report.

Point 1: Have the authors considered to analyze left vs right carotid bodies. There are evidences that the left carotid body is bigger than the right one and this may be crucial for the analysis herein performed. Also, evidences suggest differences between females vs males. Could the authors include this information in the manuscript. Please see the manuscript by Ortega-Sanz et al. 2013 J Physiol.

Response 1: Yes, we thought about comparing the left and right carotid bodies. But the technical features of performing autopsies in some clinics in our country impose some restrictions on the obtaining of postnatal carotid bodies from both sides. Therefore, we had to abandon it in this work. What about differences between females vs males. Thank you for information about manuscript by Ortega-Sanz. However, the purpose of our work was precisely the comparison of antenatal and postnatal carotid bodies, so we did not focus on the differences between male and female organs. Taking into account the fact that the work of Ortega-Sanz did not show statistically significant differences between men and women, it seems that it is more expedient to conduct such a study on a much larger sample and preferably on the same age groups.

Point 2: Sympathetic-mediated diseases have shown to change CB size in humans with the carotid bodies of patients with diabetes mellitus, hypertension, and congestive heart failure being 20-25% larger relative to controls (Cramer et al. 2014 https://pubmed.ncbi.nlm.nih.gov/24156799/). Also, the majority of the carotid bodies evaluated in the present study were from very old patients probably under drug regimens. These 2 factors might interfere with the relative expression of TH and with the ratio TH/BetaIII-tubulin.

Response 2: We also consider that such diseases as diabetes mellitus, hypertension or heart failure may influence the TH/β3 ratio. Therefore, we tried to select patients with the least number of diseases and those with the mildest disease. However, as it can be seen from our article, even in a healthy 24-year-old woman (case 22) who died from an accident, the TH/β3 ratio was significantly lower than in the antenatal period. Thus, we consider that although diseases can affect the TH/β3 ratio, this does not have much effect on the differences between the antenatal and postnatal periods of the carotid body development.

Point 3: Ortega-Sanz et al. 2013 previously reported a decrease of TH expression with age in the human carotid bodies something that was already reported in rats (Conde et al. 2006, J Neurochem https://pubmed.ncbi.nlm.nih.gov/16899065/ and Sacramento et al. 2019 https://pubmed.ncbi.nlm.nih.gov/31426127/ )and in agreement with the decline of carotid body functionality with ageing , demonstrated by decreased hypoxic responses. Probably these literature deserves to be discussed.

Response 3: We have read Ortega-Saenz's article and will certainly add their results to the discussion.

Reviewer 2 Report

In this study, Otlyga and colleagues have performed a histological characterization of the human carotid body to compare the antenatal and postnatal stages of development. The carotid body is essential to maintain oxygenation of most demanding organs such as the brain and the heart when oxygen supply is limited. Therefore, understanding how the carotid body develops and the mechanisms underlying oxygen sensing is an important question in physiology with broad applications in medical research.

This study uses extremely valuable human samples across different gestational stages to compare them with adult samples. The general quality of the micrographs and immunostaining is very good. However, the authors mostly provide a descriptive analysis of their samples and lack any functional characterisation, which I understand is difficult to obtain given the nature of the samples.  

I have some major points that may improve the general quality of the manuscript.

  1. The authors should perform a more quantitative analysis of their precious samples. The authors claim that the relative size of the carotid body in the antenatal period is larger in comparison with adjacent structures. A comparison of the number of type I cells relative to the carotid body volume or area in the antenatal versus postnatal period may be more informative.
  2. One of the main claims of the authors is the decreased expression of TH in type I cells with age. This allows them to conclude that the carotid body might play an endocrine role during the antenatal period shifting toward a chemoreceptor function later after birth. This observation is not novel (see Ortega Saenz et al., 2013 doi:10.1113/jphysiol.2013.263657)           and it should be acknowledged and extensively discussed in the discussion. Ortega et al., 2013, reported a low number of TH+ cells in the adult human carotid body that decreases with age. However, they did saw a great amount of dopa decarboxylase (DDC+) expressing glomus cells, indicating that glomus cells may still produce catecholamines in adult life. Indeed, they performed recordings of the secretory activity of the human carotid body by amperometry and observed a similar catecholamine secretion rate to those in rats. These findings should be discussed accordingly and integrated with your suggested model.

Minor points:

  1. Some arrows or asterisks may help readers to follow the description of the histological features in the text.

Author Response

Thank you for your Review Report.

Point 1: The authors should perform a more quantitative analysis of their precious samples. The authors claim that the relative size of the carotid body in the antenatal period is larger in comparison with adjacent structures. A comparison of the number of type I cells relative to the carotid body volume or area in the antenatal versus postnatal period may be more informative.

Response 1: We give the relative sizes of the organ only for case 1, since in this case the embryo was investigated totally on serial sections, which made it possible to compare the structures directly histologically. Unfortunately, for other samples, an accurate morphometric study on histological sections is impossible, since the carotid bodies were dissected from the surrounding tissues. In these cases, a macroscopic examination of the size will be inaccurate, since the size of the carotid body will be different depending on the degree of ‘cleaning’ during the dissection of the organ, which Ortega also writes about in the article you cited.

Point 2: One of the main claims of the authors is the decreased expression of TH in type I cells with age. This allows them to conclude that the carotid body might play an endocrine role during the antenatal period shifting toward a chemoreceptor function later after birth. This observation is not novel (see Ortega Saenz et al., 2013 doi:10.1113/jphysiol.2013.263657)           and it should be acknowledged and extensively discussed in the discussion. Ortega et al., 2013, reported a low number of TH+ cells in the adult human carotid body that decreases with age. However, they did saw a great amount of dopa decarboxylase (DDC+) expressing glomus cells, indicating that glomus cells may still produce catecholamines in adult life. Indeed, they performed recordings of the secretory activity of the human carotid body by amperometry and observed a similar catecholamine secretion rate to those in rats. These findings should be discussed accordingly and integrated with your suggested model.

Response 2: Thank you very much for the information on Ortega-Saenz's article. However, we mainly focused on the differences between the carotid body in the antenatal and postnatal periods of development, whereas Ortega-Saenz examined the changes that occur in postnatal development. But despite this, the Ortega-Saenz's data is very interesting and we will certainly include it in the discussion in our article.

Point 3: Some arrows or asterisks may help readers to follow the description of the histological features in the text.

Response 3: We will add these elements to the micrographs.

Round 2

Reviewer 1 Report

The authors addressed my comments.

Reviewer 2 Report

No further comments.